# Influences of Dietary Supplementation with Maca (*Lepidium meyenii*) on Performance, Parameters of Growth Curve and Carcass Characteristics in Japanese Quail

**DOI:** 10.3390/ani12030318

**Published:** 2022-01-28

**Authors:** Firdevs Korkmaz Turgud, Doğan Narinç

**Affiliations:** Department of Animal Sciences, Akdeniz University, Antalya 07100, Turkey; dnarinc@akdeniz.edu.tr

**Keywords:** maca, gompertz growth curve, feed additives, chick quality, quail

## Abstract

**Simple Summary:**

It is known that the bioactive compounds (*N*-benzyl-palmitamide, benzyl isothiocyanate, glucosinolates and phenolics) in the maca plant are appetizing, as well as having antioxidant effects and improving reproductive ability. The aim of this study was to determine the effects of adding maca plant powder to the ration at different levels on growth, slaughter carcass, partial egg production and some reproductive characteristics of Japanese quail (*Coturnix coturnix japonica*). Addition of maca powder to the diet increased the feed consumption but did not affect growth, slaughter carcass, partial egg production or fertility. In addition, maca powder reduced embryonic deaths and improved chick quality. According to these results, the positive effects of short-term application of maca powder are not observed during the fattening period in Japanese quails.

**Abstract:**

Maca plant contains rich nutrients and in addition, it has various bioactive substances (*N*-benzyl-palmitamide, benzyl isothiocyanate, glucosinolates and phenolics). It is used to improve reproductive properties and has antioxidant effects for both humans and animals. The aim of this study was to determine the effects of adding maca plant powder to the ration at different levels on growth, slaughter carcass, partial egg production and some reproductive characteristics in Japanese quail (*Coturnix coturnix japonica*). The experimental groups were formed by adding 0% (control), 0.05% and 0.1% maca powder to the diet, and a total of 300 birds were used. Growth (weekly body weights, parameters of Gompertz growth function), feed efficiency and carcass characteristics of quails in the 42-day fattening trial were determined. Reproductive characteristics were measured up to 22 weeks of age. Addition of maca powder to the diet increased the feed consumption (*p* < 0.05) but did not affect body weights at 35 and 42 days of age, β_0_ and β_1_ parameters or point of inflection weight of the Gompertz model, carcass traits, partial egg production or fertility. It may be advisable to add 0.1% maca powder to the diets of breeders. Besides, maca powder reduced embryonic deaths and improved chick quality (both *p* < 0.05). It is thought that different results for reproductive traits can be obtained if maca powder is used for a longer period in the diets of breeder quail flocks.

## 1. Introduction

Antibiotic (tetracyclines, avoparcin, virginiamycin, tylosine, spiramycin, etc.) use, which aims to increase the digestion level by controlling the intestinal microbiota in chicks, is a common practice. Thus, feed efficiency is improved, and more live weight gain can be achieved. However, this practice, which is considered risky for public health, has been banned in European Union countries since 2006. For this reason, new natural or biotechnological feed additives are being researched in order to create the effect of antibiotics in poultry. Plants have been used for therapeutic purposes since prehistoric times. In the structure of each plant, there are many important bioactive compounds such as alkaloids, flavenoids, glycosides, mucilages, saponins, tannins, and phenol, phenolic acids, coumarin, terpenes, essential oils, lectins and polypeptides [1]. These chemical compounds in plants show antiviral and antibacterial effects in other living things that consume plants and cause many effects that strengthen the immune system. Bioactive compounds such as *N*-benzyl-palmitamide, benzylisothiocyanate, glucosinolates and phenolics in maca have important effects on lipid, mineral and antioxidant metabolisms [2,3,4,5]. Many plants and their extracts as feed additives in animal nutrition have been used for many years for growth, reproduction, improvement of product quality and health protection.

Maca (*Lepidium meyenii*) is a medicinal plant with high nutritional value (12.8% protein in dry matter content, 23.5% fibrous substance) from the Brassicaceae family and is a plant from the Andes of Peru. Maca plant contains important fatty acids, macacids, macaridin, alkaloids and glucosinolates [6]. In addition, maca plant has an aphrodisiac compound called p-methoxybenzyl isothiocyanate [7]. Researchers have focused on the effects of using a powder or extract of maca plant as a food additive for humans on some reproductive traits in men and women [8,9]. In addition, it was reported that bone mineralization against osteoporosis and antioxidant activity against stress were improved in rats consuming extracts obtained from the maca plant [2,10]. There are limited studies on the use of maca as a feed additive in livestock diets. In the study conducted by Clément [11], it was determined that maca powder given to breeding cows as a feed additive increased their sperm quality and number. A similar study was carried out in sheep, and it was determined that maca powder given to breeders as a feed additive increased the number of copulation and ejaculation [12]. Bilal et al. [13] determined that the addition of 50 and 75 g/day of maca powder to racehorse rations had no effect on body weight and average daily feed consumption characteristics. In a study by El-Sheikh et al. [14], the addition of maca to the diet did not have a significant effect on the live weight, daily feed consumption, litter size, litter weight or milk production of rabbits. To our knowledge, the only study examining the effects of adding maca to poultry diets was conducted by Korkmaz et al. [15]. The researchers investigated the effects of maca powder supplementation on the performance, egg quality, serum parameters, hormones and antioxidant enzyme levels of laying hens in the post-peak period. In their study, maca powder had neither positive nor negative effects on performance, egg quality, egg yolk cholesterol content, serum parameters (excluding magnesium) or hormones. The aim of this study was to determine the effects of adding maca plant powder to the ration at different levels on growth, slaughter carcass, egg production and some reproductive characteristics of Japanese quail (*Coturnix coturnix japonica*). Among avian species, Japanese quail has been used in many studies related to early sexual development and maturation [16,17]), as well as for the high production of the eggs and their regular deposition [18].

## 2. Materials and Methods

### 2.1. Animal Material

The study was carried out in poultry facilities of the Faculty of Veterinary Medicine and the Faculty of Agriculture of Namık Kemal University, based on the decision of Namık Kemal University Animal Experiments Local Ethics Committee, dated 18 October 2017 and numbered 78. The animal material of the study consisted of a total of 300 quail chicks, which were randomly mated and obtained simultaneously from a parent flock that had not been genetically selected before.

### 2.2. Rearing Period

One-day-old chicks were randomly assigned to each experimental group (100 birds per group), and wing numbers were assigned immediately after hatching, and individual weights were recorded. Weekly live-weight measurements were repeated individually thanks to the wing numbers. The chicks, which were randomly allocated to each experimental group, were fed in brooder cages (battery type with 5 floors, 90 cm^2^/quail) until sex determination on the 21st day after hatching. The chicks were housed at 32 °C for the first three days, and this was lowered by 1 °C every three days, and the temperature was adjusted to 27 °C at the end of the second week. After the 21st day, the quails were housed as a group in the rearing cages (battery type with 5 floors, 160 cm^2^/quail). A constant light intensity (60 lux) was used throughout the study. A lighting program of 23 h of light and 1 h of darkness was applied to all quails during the first six weeks.

Experimental groups were formed by adding 0% (control; C), 0.05% (M1) and 0.1% (M2) maca powder to the diet. Three different rations were used throughout the experiment, and these are presented in Table 1 according to the periods in which they were used. In the study, feed consumption was determined as a group in brooder cages (from hatching to 3 weeks of age). Then, it was determined individually in the rearing period (between 3 and 6 weeks of age). 

The Gompertz equation was fitted to the growth data of Japanese quails to model the relationship between body weight and age [19]. The model expression of the Gompertz function and their coordinates of the point of inflection are presented in Table 2.

In the equation, “t” denotes time, “Y” weight, “β_0_” the maximum body weight the animal is assumed to be able to reach, “β_1_” the biological constant of the shape of the curve, “β_2_” the biological of about the growth rate and “β_3_” the shape parameter [20]. Model parameters were analyzed using with SAS 9.3 software NLIN procedure Levenberg–Marquardt iteration method [21,22]. Fifty randomly selected quails from each experimental group and a total of 150 quails (71 females and 79 males) were sent to slaughter at the age of 42 days. Feed was removed for 4 h before slaughter, and slaughter weights of quails were determined. All weight measurements during cutting were carried out with a digital scale with 0.01 kg precision. Following slaughter, wet plucking and evisceration, hot-carcass weights were determined, including neck and belly fat, excluding edible internal organs. At this stage, edible visceral weights consisting of abdominal fat, heart, liver and empty gizzard were determined [23]. After the carcasses were kept at +4 °C for one day, the cold carcass weight was measured, and the carcasses were shredded, and the breast weight, chest muscle weight, thigh and wing weights were measured. By dividing the cold carcass, edible internal organs, abdominal fat, breast, thigh and wing weights to the slaughter weight, phenotypic values were obtained for cold carcass ratio, edible internal organs ratio, abdominal fat ratio, breast ratio, breast muscle ratio, thigh ratio and wing ratio, respectively [5,24].

### 2.3. Reproductive Period

Of the quails that were not selected for slaughter, 27 females and 9 males (1:3 mating ratio) from each group were transferred to individual breeding cages to determine egg production and some reproductive characteristics (fertility, embryonic mortalities, chick quality). We aimed to ensure high fertility by placing the male quail of each family in the cage of only one of the three females every day [25]. Hen-housed day egg yields were individually kept up to 22 weeks of age. When the quails were 10, 14, 18 and 22 weeks old, the eggs collected for 5 days were put into the incubator daily, and the hatching results were evaluated. For this purpose, fertility, total embryonic death, early embryonic death and late embryonic death were determined by macroscopic examination. All hatched chicks were examined by experienced operators to determine the Tona chick quality score of chicks as previously described by Tona et al. [26]. The Tona scoring method is a qualitative scoring system that assesses a total score index of 100, based on a wide variety of visual parameters, such as activity, appearance, retracted yolk, eye condition, leg and feet condition, navel deformities, and status, remaining egg membrane, beak condition and remaining yolk [27].

### 2.4. Statistics

To determine the differences between the means of the groups in terms of all the characteristics obtained, analysis of variance was applied. Duncan multiple range test was applied when the H_1_ hypothesis was accepted. The significance level was accepted as 0.05 in all statistical analyses. All statistical analyses were performed using SAS 9.3 statistical software.

## 3. Results

The average values of some performance characteristics of female and male quails at 35 and 42 days of age and the results of the variance analysis are given in Table 3. As can be seen from Table 3, the absence of a statistical difference between hatching weights of quails indicates that the subjects were randomly assigned to the experimental groups. It was determined that the quails in the M1 and M2 groups did not have different averages from birds in the control group in terms of body weight and feed conversion ratio at 5 and 6 weeks of age (*p* > 0.05 for both traits and weeks). However, it was determined that adding maca powder to the diet increased feed consumption linearly (*p* < 0.05). In the study, statistical differences were determined between the sexes in terms of mean values of body weight and the feed conversion ratio. While female quails had higher body weights than males, they also had a better feed conversion ratio. The treatment–sex interaction effect was not found to be significant in terms of body weight, feed consumption or feed conversion ratio (*p* > 0.05 for all).

The results of the Gompertz growth curve analyses performed using the weekly live-weight data of female and male quails in the experimental groups are given in Table 4. In addition, the growth curves drawn according to the experimental groups and genders are also presented in Figure 1. The coefficients of determination in all nonlinear regression analyses were found to be between 0.996 and 0.999 (not included in any table). In this case, it was determined that the Gompertz growth curve model was quite sufficient to explain the quail data. The mean values of the mature (asymptotic)-weight parameter (β_0_) of the Gompertz growth model were estimated as 241.57, 258.63 and 260.23 g for the C, M1 and M2 groups, respectively (Table 4). The mean β_0_ value of the control group was not statistically different from those in the M1 and M2 groups (*p* > 0.05). At the same time, the mean value of the β_0_ parameter of female quails was higher than that of males (*p* < 0.05). There was no difference between the experimental groups (*p* > 0.05) and between the sexes (*p* < 0.05) for the β_1_ parameter, which is defined as the integration constant or maturation rate. The highest mean value (3.90) for the β_1_ parameter was found in females in the M2 group, while the lowest mean value (3.56) was determined in males in the control group. There was a significant difference between the experimental groups in terms of growth rate parameter (β_2_), and the β_2_ values of quails given maca powder were lower than those of group C (*p* < 0.05). It can be seen that quails in the control group reached the point of inflection of the Gompertz growth model at an earlier age (*p* < 0.05), but there was no difference between the experimental groups in terms of the weight of the point of inflection. Contrary to this situation, there was no difference between the time of the point of inflection of female and male quails, while females reaching the point of inflection were found to be heavier (*p* < 0.05). The inflection point age of the Gompertz model was between 16.61 and 19.79 days in all groups. The highest point of inflection weight average was found in the female quails of the M2 group (101.68 g), while the lowest mean value of point of inflection weight was found in the control group male quails (83.59 g).

The mean values determined for carcass yield and breast, breast meat, leg, wing and abdominal fat ratios (% body weight) of quails slaughtered at the age of six weeks in the control, M1 and M2 experimental groups, as well as variance analysis results, are presented in Table 5. The addition of maca powder to the ration did not affect any carcass characteristics. Similarly, no difference in carcass characteristics was observed between males and females. Carcass yields of quails varied between 67.46 and 71.74%.

The mean values of egg yield, fertility, early, late and total embryonic mortality and the analysis of variance are presented in Table 6. There was no statistical difference between the experimental groups in terms of egg yield and fertility. However, the mean values of the groups fed with maca powder were found to be lower in terms of embryonic deaths. While the mortality rate of quails in the control group was 11.26%, embryonic mortality in the M1 and M2 groups was 8.98% and 4.89%, respectively (*p* < 0.05). A similar situation occurred in terms of chick quality, the Tona score of quails fed with 0.1% maca powder added to the ration had the highest mean value (*p* < 0.05). The mean values of Tona chick quality scores in the C, M1 and M2 experimental groups were determined as 90.67, 89.44 and 95.42.

## 4. Discussion

In the study, it was determined that the addition of 0.05% and 0.1% maca powder to the diet had no effect on body weight and feed efficiency at 35 and 42 days of age but negatively affected the amount of feed consumption (*p* < 0.05). It was determined that the quails in the M2 group consumed higher amounts of feed (594.78 g and 781.93 g) at 35 and 42 days of age compared to the other groups. It has been determined that the effect of adding maca powder to the feed consumption increases linearly with age. Maca powder is thought to have appetizing properties, and it can be seen that the addition of 0.05% and 0.1% maca powder, especially at the age of six weeks, significantly increased the amount of feed consumption compared to the control group. There is only one study on the addition of maca powder to the diets of poultry. In the mentioned study (Korkmaz et al., 2016) it was determined that the addition of 5 and 10 g/kg maca powder to the rations of laying hens at 56 and 72 weeks of age did not affect the live weight and feed efficiency. The results of this study are consistent with the results reported by Korkmaz et al. [15]. In another study conducted in ruminants, it was reported that the addition of maca powder to the ration did not affect the feed intake, growth rate or carcass performance of bulls [28]. In the study performed by Uchiyama et al. [29], it was determined that the growth rates and feed intakes of rats fed with diets with three different doses of maca powder for seven weeks were not affected by maca supplementation.

It was determined that the gender effect influenced the body weight and feed efficiency characteristics of quails (*p* < 0.05). According to the body weights at 35 and 42 days of age, higher values were found for females (187.99 g and 211.40 g, respectively) than males (177.00 g and 199.51 g). Although female and male quails consumed similar amounts of feed, it was determined that females with a higher body weight also had better feed conversion ratios (Table 2). There is reverse dimorphism between males and females in Japanese quail compared to other poultry species. In many studies, it has been reported that weekly live-weight values and feed consumption characteristics of female quails are higher than males [30].

According to our knowledge, there is no study on the effect of adding maca powder to poultry rations on growth curve parameters. There are few studies on the effects of feeding poultry with different rations on growth curves [20]. Lilburn et al. [31], who raised two different turkey genotypes (Nicholas and British United Turkeys) using two lighting programs and two feeding programs, compared the growth patterns of the experimental groups with the Gompertz model. There was no significant dietary effect on any growth curve parameter in their study [31]. Taroco et al. [32] investigated the genotype–environment interaction effects of adding five different levels of threonine:lysine to Japanese quail diets in terms of Gompertz model parameters. Researchers reported that this application had significant effects on the heritability and phenotypic values of the asymptotic weight parameter and inflection point age characteristics of the Gompertz model. Hashiguchi and Yamamoto [33], who examined the growth of Japanese quails fed with diets containing different ratios of protein with the Gompertz model, reported that this practice affected the growth curve parameters. The mature-weight parameter values (140.3–156.4) estimated by the researchers were found to be considerably lower than the averages determined in this study. It is thought that the reason for this situation is due to the low weekly live weight values of the quails used in the study by Hashiguchi and Yamamoto [33]. There are quite a few differences between the live weight values of Japanese quails due to the domestication, adaptation to cage conditions and genetic selection studies. While body weight values at the age of six weeks have been reported as 100–130 g in some studies [34,35], these averages have been reported to be in the range of 250–300 g in other studies [36,37]. In studies in which the growth of Japanese quails was examined with the Gompertz model, the mature-weight parameter was found in the range of 224–295 g [38,39]. In the study, β_0_ parameter averages (241.57–260.23 g) obtained from all three experimental groups were found to be compatible with mature-weight parameter values reported in the literature. In the study, the integration coefficient parameter (β_1_) of the Gompertz growth curve model for quail growth samples was estimated in the range of 3.60–3.83. The results obtained were found to be compatible with the estimated values (3.40–3.89) for randomly mating herds that were not selected by many researchers [39,40,41]. In the study, it was determined that adding maca to the diet decreased the instantaneous growth rate parameter; therefore, the birds reached the inflection point of the sigmoid growth curve in the later period. It is thought that this situation is caused by the effect of increasing feed consumption. Small values for β_2_ indicate late maturation and a high adult weight. On the other hand, high β_2_ values represent early maturation and a lower adult weight [40]. It is thought that if the maca dose in the ration is increased, the mature-weight parameter will also be positively affected by this situation. In all experimental groups in the study, the inflection point age of the Gompertz model was between 17.08 and 19.01 days, and the mean weight of the inflection point was between 88.87 and 95.73 g. According to the results of many studies in which the growth samples of Japanese quails were analyzed with the Gompertz function, it was reported that the values obtained for the inflection point age of the curve were between 14.76 and 24.62 days of age, and the weight of the growth curve inflection point was between 76.22 and 124.56 g [19,38,39,42,43,44]. The values of inflection point age and weight determined for the Japanese quails included in this study were found to be consistent with the averages reported in these studies.

It was determined that the addition of maca powder to quail rations did not cause any difference in terms of yield of carcass, breast, breast meat, thigh, wing or abdominal fat (Table 5). The maca powder used in the study did not affect the weekly live weight or slaughter weight and did not affect the slaughter or carcass characteristics as expected. Ginseng, ginger and licorice root are medicinal plants that contain phenolic and bioactive components similar to maca. These plants have also been used as additives in poultry feeds for years. Reda et al. [45] investigated the effects of 250, 500, 750 and 1000 mg/kg of licorice root supplementation on the performance of Japanese quails. In parallel with our results, researchers reported that the doses applied did not have any effect on the carcass. On the other hand, it was stated that abdominal fat decreased in broiler chickens when 0.3 g/L of licorice root was added to drinking water and 2 g/kg of licorice root was included in feed [46]. Azazi et al. [47] reported that adding different levels of ginseng to rations of a layer breeder flock improved semen quality, fertility and hatchability. Researchers reported that adding ginseng to the diet did not affect percentages of edible inner organs, but surprisingly increased carcass yield. It is thought that the reasons for this difference are that the animal material used is old and that ginseng has been applied for a long time. In accordance with the results of this study, many researchers [48,49,50,51,52] reported that the carcass yields of Japanese quail were between 68.33 and 73.00%. Having obtained similar results, Walite et al. [53] reported that the mean values of the breast, leg and wing were 29.1%, 15.9% and 11.8%, respectively. Akbarnejad et al. [54] found lower averages than the mean values determined in this study, and the yield of cold carcass, breast and leg were found to be 65.0%, 24.7% and 15.1%, respectively. Egg production, fertility, early, late and total embryonic death rates and mean analysis of variance are given in Table 6. While there was no statistical difference between the experimental groups in terms of egg production and fertility, the average values of the groups fed with maca powder were found to be lower in terms of embryonic deaths. The alkaloids present in the maca plant work as the main stimulant of the ovarian follicles. It also has the effect of eliminating free radicals and has an antioxidant function; antioxidants protect the body against their harmful effects by neutralizing free radicals that are released as by-products during cellular metabolism and energy production in mitochondria. Antioxidants may be a mechanism of increasing fertility by enhancing LH levels [55]. 

It is a known fact that the use of plant-derived antioxidants improves reproductive traits such as quality of the sperm, semen, oocyte and embryo. It was determined that the addition of 5 and 10 g/kg maca to the laying hens’ rations did not have any positive or negative effect on the egg production performance [15]. The results of the study performed by Korkmaz et al. [15] are consistent with our findings. However, Osfor [56] reported that adding 2 and 4 mg/kg of ginseng plant, which has similar antioxidant effects to maca, to quail rations increases egg production and weight and improves feed efficiency. Similarly, Jang et al. [57] reported that the addition of fermented ginseng by-product to laying hens’ rations increased egg weight and yield. Al-Kassie [58] stated that cumin used at different levels in broiler rations causes significant differences in mortality rates; the lowest mortality rate was determined at the level of 0.5% and 1 cumin supplementation (4.1–3.4%) to the ration. However, Ebrahimi et al. [59] reported that adding medicinal and aromatic plants to broiler rations did not affect mortality rates. It was concluded that the addition of some plant seeds and root extracts with antioxidant effects to the rations of breeder turkey hens improves fertility, hatchability and embryonic viability [60,61].

## 5. Conclusions

Maca plant had no effect on body weight or carcass characteristics of quails. The most important positive effects in the study are related to total embryonic death and chick quality characteristics. It may be advisable to add 0.1% maca powder to the diets of breeder flocks. It is claimed that the use of plant-based antioxidants in livestock has a positive effect on reproductive characteristics, but compatible results were not obtained in our study. This study was carried out during a short interval of the laying period. For this reason, it is thought that different results for reproductive traits can be obtained if maca powder is used for a longer period in the diets of breeder quail flocks. These results show that the effects of maca plant on reproductive traits in Japanese quail should be investigated in future studies.

## Figures and Tables

**Figure 1 animals-12-00318-f001:**
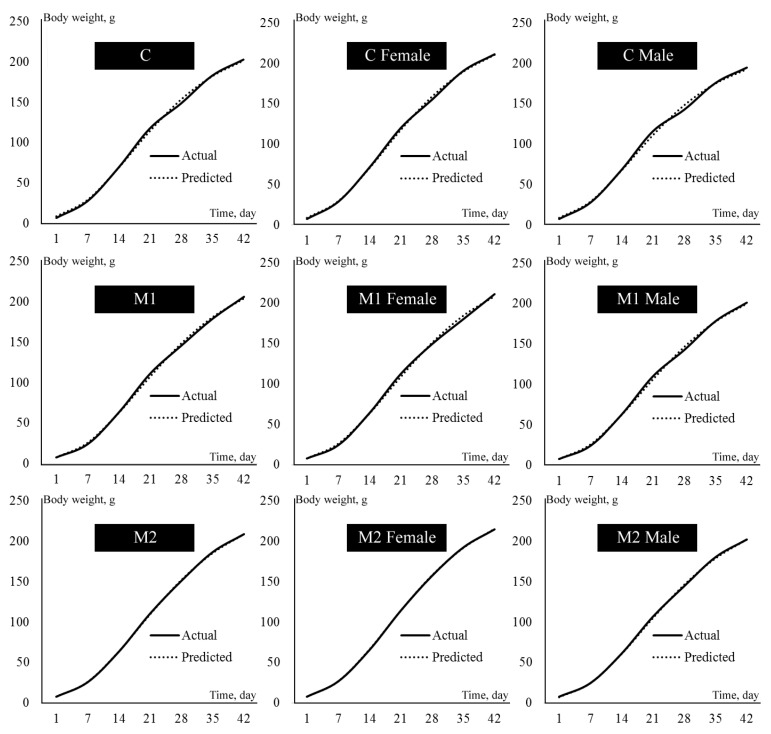
Actual and predicted growth curves by treatment for groups and sex.

**Table 1 animals-12-00318-t001:** The nutritional compositions of diets for the chick rearing period (0–14 days), growing period (15–42 days) and egg-laying period (42 days and after) of quails.

Ingredients	Starter (0–14 Days)	Grower (15–42 Days)	Breeder (42 and Later Days)
Corn	51.5	58.50	54.17
Soybean meal	41.5	36.00	34.70
Vegetable oil	3	1.5	1.11
Limestone	1.25	1.25	7.01
Dicalcium phosphate	1.6	1.6	1.15
Salt	0.35	0.35	0.36
d-L Methionine	0.30	0.30	-
l- Lysine	0.15	0.15	-
vit-min premix ^1,2,3^	0.50	0.50	1.50
Maca powder	0–5–10 g/kg	0–5–10 mg/kg	0–5–10 g/kg
Calculated Values			
ME	2910	2900	2800
CP	24	22	19.46
Calcium	0.98	0.96	3.07
Available *p*	0.42	0.41	1.31

^1^ Vitamin and mineral mix per kg of ration: retinol acetate 1706 mg, cholecalciferol 41 mg, DL-αtocopherol 27 mg, menadione 0.99 mg, cobalamin 0.015 mg, folic acid 0.8 mg, D-pantothenic acid 15 mg, riboflavin 5.4 mg, niacin 45 mg, thiamine 2.7 mg, D-biotin 0.07 mg, pyridoxine 5.3 mg, manganese 90 mg, zinc 83 mg, iron 121 mg, copper 12 mg, iodine 0.5 mg, selenium 0.3 mg. ^2^ Digestarom® poultry, herbal extract blend 100 mg/kg. ^3^ Laying core composition/kg of the product: calcium (min) 80 g/kg (8%), calcium (max) 100 g/kg (10%), phosphorus (min) 37 g/kg (3.7%), sodium (min) 20 g/kg, methionine (min) 21.5 g/kg, lysine (min) 18 g/kg, vitamin a (min) 125,000 IU/kg, vitamin D3 (min) 25,000 uu/kg, vitamin E (min) 312 UI/kg, vitamin k3 (min) 20 mg/kg, vitamin B1 (min) 20 mg/kg, vitamin B2 (min) 62.5 mg/kg, vitamin B6 (min) 37.5 mg/kg, vitamin B12 (min) 200 mcg/kg, folic acid (min) 6.25 mg/kg, pantothenic acid (min) 125 mg/kg, biotin (min) 1.25 mg/kg, choline (min) 1700 mg/kg, niacin (min) 312 mg/kg, copper (min) 125 mg/kg, iron (min) 68 0 mg/kg, iodo (min) 8.75 mg/kg, manganese (min) 937 mg/kg, selenium (min) 3.75 mg/kg, zinc (min) 500 mg/kg, fluoride (max) 370 mg/kg.

**Table 2 animals-12-00318-t002:** Expression and point of inflection coordinates of Gompertz growth function.

Growth Model	Equation	Inflection Point Age	Inflection Point Weight
Gompertz	$Y_{t}{=\beta }_{0}\cdot e^{-\beta_{1}e^{-\beta_{2}t}}$	ln(β_1_)/β_2_	β_0_/e

**Table 3 animals-12-00318-t003:** The mean values of some performance traits and results of variance analyses.

Effects		Hatch Weight	BW 35	BW 42	FC 35	FC 42	FCR 35	FCR 42
Treatment								
C		7.45	183.05	202.78	581.33 ^b^	768.35 ^c^	3.34	3.98
M1		7.64	178.56	205.77	588.58 ^b^	779.44 ^b^	3.51	4.03
M2		7.64	185.86	207.81	614.20 ^a^	798.17 ^a^	3.51	4.07
Sex								
F		7.61	187.99 ^a^	211.40 ^a^	594.63	782.04	3.35 ^b^	3.93 ^b^
M		7.54	177.00 ^b^	199.51 ^b^	594.78	781.93	3.56 ^a^	4.12 ^a^
Interaction								
C	F	7.57	190.02	210.87	581.78	768.76	3.21	3.83
	M	7.34	176.09	194.70	580.88	767.94	3.47	4.14
M1	F	7.63	183.63	209.90	587.36	777.69	3.39	3.98
	M	7.64	173.50	201.64	589.80	781.18	3.63	4.09
M2	F	7.63	190.31	213.44	614.75	799.67	3.45	3.99
	M	7.64	181.40	202.18	613.66	796.66	3.57	4.15
SEM		0.06	1.92	2.44	1.54	2.28	0.04	0.05
Variation Source		*p* Values						
Treatment		0.356	0.294	0.702	0.000 *	0.000 *	0.123	0.787
Sex		0.557	0.005 *	0.016 *	0.961	0.981	0.008 *	0.048 *
Trt–Sex		0.650	0.857	0.800	0.870	0.839	0.756	0.692

F: female, M: male, BW: body weight, FC: feed consumption, FCR: feed conversion ratio, Trt: treatment. The means with different letters in the same column for the subgroups of main effect or interaction are statistically different, * *p* < 0.05.

**Table 4 animals-12-00318-t004:** The mean values of Gompertz growth curve parameters and results of variance analyses.

Effects		β_0_	β_1_	β_2_	IPT	IPW
Treatment						
C		241.57	3.60	0.077 ^a^	17.08 ^b^	88.87
M1		258.63	3.77	0.073 ^b^	19.01 ^a^	95.14
M2		260.23	3.83	0.073 ^b^	18.86 ^a^	95.73
Sex						
F		268.91	3.78 ^a^	0.073	19.03	98.93 ^a^
M		238.05	3.68 ^b^	0.076	17.60	87.57 ^b^
Interaction						
C	F	255.95	3.64 ^d^	0.077	17.55	94.17
	M	227.19	3.56 ^e^	0.078	16.61	83.59
M1	F	274.42	3.80 ^b^	0.072	19.74	100.96
	M	242.83	3.75 ^c^	0.074	18.27	89.34
M2	F	276.35	3.90 ^a^	0.071	19.79	101.68
	M	244.11	3.75 ^c^	0.075	17.92	89.81
SEM		5.32	0.06	0.001	0.32	0.92
Variation Source		*p* Values				
Treatment		0.356	0.289	0.001 *	0.029 *	0.356
Sex		0.037 *	0.004 *	0.074	0.335	0.037 *
Trt–Sex		0.650	0.016 *	0.773	0.932	0.650

F: female, M: male, BW: body weight, FC: feed consumption, FCR: feed conversion ratio, Trt: treatment, β_0–2_: model parameters of Gompertz exuqtion, IPT: point of inflection time, IPW: point of inflection weight. The means with different letters in the same column for the subgroups of main effect or interaction are statistically different, * *p* < 0.05.

**Table 5 animals-12-00318-t005:** The mean values of the carcass characteristics (%) and results of variance analyses.

Effects		Carcass Yield	Abdominal Fat	Breast	Breast Meat	Leg	Wing
Treatment							
C		71.27	0.47	28.95	18.98	16.31	6.01
M1		68.32	0.39	27.46	18.47	15.47	6.14
M2		71.41	0.43	29.03	19.62	16.14	6.04
Sex							
F		70.70	0.44	28.51	19.06	16.14	6.12
M		69.97	0.42	28.46	18.99	15.81	6.01
Interaction							
C	F	71.74	0.44	29.38	19.52	16.51	6.08
	M	70.81	0.50	28.53	18.43	16.11	5.93
M1	F	69.18	0.45	27.66	18.25	15.54	6.19
	M	67.46	0.32	27.25	18.69	15.40	6.10
M2	F	71.18	0.43	28.48	19.41	16.36	6.09
	M	71.64	0.43	29.58	19.84	15.92	5.99
SEM		0.06	0.67	0.04	0.35	0.34	0.21
Variation Source		*p* Values					
Treatment		0.356	0.119	0.730	0.123	0.394	0.230
Sex		0.557	0.590	0.785	0.943	0.916	0.437
Trt–Sex		0.650	0.801	0.651	0.488	0.569	0.950

F: female, M: male.

**Table 6 animals-12-00318-t006:** The mean values of egg yield, reproductive characteristics, chick quality and results of variance analyses.

Treatment	Egg Yield (%)	Fertility (%)	Total Embryonic Mortality (%)	Early Embryonic Mortality (%)	Late Embryonic Mortality (%)	Tona Score
C	90.70	86.99	11.26 ^a^	5.16	6.11	90.67 ^b^
M1	88.99	85.38	8.98 ^a^	3.31	5.67	89.44 ^b^
M2	87.84	84.28	4.89 ^b^	2.73	2.17	95.42 ^a^
SEM	3.26	2.90	0.85	0.56	0.70	1.13
*p* values	0.618	0.533	0.025 *	0.216	0.072	0.042 *

The means with different letters in the same column are statistically different, * *p* < 0.05.

## Data Availability

The data presented in this study are available on request from the corresponding author.

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
