# Peer review of "Influences of Dietary Supplementation with Maca (Lepidium meyenii) on Performance, Parameters of Growth Curve and Carcass Characteristics in Japanese Quail"

_animals, 2022, doi:10.3390/ani12030318_

Round 1

Reviewer 1 Report

It is necessary to address the following recommendations to the manuscript:

Line 12: scientific names must be written in italic text format.

Line 30: according to the Microsoft word template included in the authors guide, the first letter of each keyword must be written in lowercase text format.

Line 42: according to the Microsoft word template included in the authors guide, the references must be cited in numeric format between brackets.

Line 40: did you mean flavonoids and bioflavonoids?

Line 41: terpene or terpenes?

Line 33: Could you mention some examples of antibiotics used?

Line 50: indicate which region the plant is native.

Line 85: n = 100 per each group?

Line 93: What was the intensity of the light used during the program (lumens)?

Line 93: indicate the dimensions of the cages used in the experiment.

Line 102: in the numerical values of the table 1, use a point instead a comma.

Line 102: according to the Microsoft word template included in the authors guide, the titles of each column in the tables, must be marked in bold format. Modify in all tables of the manuscript.

Line 103: insert space (14 mg).

Line 103: insert space (5.4 mg).

Line 105: ; manganese.

Line 107-113: insert space (80 g/kg, 100 g, 37 g, 20 g, 21.5 g, 18 g, 20 mg, 62.5 mg, 37.5 mg , etc. In addition, change the text formatting to lowercase of each component.

Line 80-159: indicate the subtitle of each experimental stage.

Line 217: insert a point at the end of the figure title.

Line 326-328: What could be the reason for these results?

Line 334: 3 g, 2 g (idem).

Line 347:  delete space between paragraphs.

Line 356: delete the point after the word levels.

Line 375: according to "Some enzymes and vitamins in maca root can create an appetizing effect", How could you conclude this statement, if the study did not determine the enzymes and / or vitamins of the maca powder?

Line 379: according to "Thanks to the phenolic compounds in maca content, it is a plant with high antioxidant capacity", How could you conclude this statement, if the study did not determine the phenolic composition of the maca powder?.

Line 380-382: according to "The use of plant-based antioxidants has become widespread in recent 380 years in order to improve reproductive characteristics such as sperm, semen, oocyte and embryo in livestock", this statement is not a conclusion of the work, it is a background statement which could be used as a discussion of results.

Line 375-386: it is necessary to modify the conclusions, and these must be based on the results obtained.

References section

According to the Microsoft word template included in the authors guide, the names of the journals in each reference must be abbreviated, use a long hyphen instead of a short hyphen to indicate the page range.

Line 457: scientific names must be written in italic text format.

Author Response

Dear reviewer, thank you for your contribution, suggestions, and criticisms to the article. All corrections have been made and are shown below.

Line 12: Scientific name italicized

Line 30: First letters of keywords converted to lowercase

Line 42: Reference notations in the text have been changed to numbers.

Line 40: bioflavenoids deleted from text

Line 41: it was changed to terpenes.

Line 34: Examples of antibiotics include "tetracyclines, avoparcin, virginiamycin, tylosine, spiramycin, etc." phrase added to the text.

Line 50 (new 53): Added information about the origin of the plant. "... and is a plant from the Andes of Peru".

Line 85 (new 88): The chicks in each experimental group were selected by chance, and each of them included 100 individuals. A statement describing this situation has been added. "One day old chicks were randomly assigned to each experimental group (100 birds per group), ..."

Line 93 (new 96): A explanation was added to text "A constant light intensity (60 lux) was used throughout the study."

Line 93 (new 90-96): İnformations about cage types and stocking density have been added. "The chicks, which were randomly allocated to each experimental group, were fed in heat-ed rearingbrooder cages (battery type with 5 floors, 90 cm2/quail) until sex determination made on the 21st day after hatching. The chicks were housed at 32 °C for the first three days and lowered by 1 °C every three days, and the temperature was adjusted to 27 °C at the end of the second week. After the 21st day, the quails were housed as a group in the breeding rearing cages (battery type with 5 floors, 160 cm2/quail)."

Line 102 (new 105): Corrections were made.

Line 102 (new 105): Column headings in all tables have been updated to bold.

Line 103-107 (new 107-119): The punctuations "," and ";" in the explanations below the Table 1 have been updated.

Line 107-113 (new 112-119): Necessary corrections have been made in the explanations below the Table 1.

Line 80-159 (new 81-170): Four subtitles (2.1. Animal Material, 2.2. Rearing Period, 2.3. Reproductive Period, 2.4. Statistics) have been added to the Material and methods section.

Line 217: Dots have been added to all table and figure captions.

Line 326-328 (new 339-340): Explanation was added to text. "Maca powder used in the study did not affect the weekly live weight and slaughter weight, and did not affect the slaughter and carcass characteristics as expected."

Line 334-335 (new 346-347): Necessary explanation has been added. "... in broiler chickens with 0.3 g/l of licorice root added to drinking water and 2 g/kg of licorice root included in feed (Alagawany et al. 2019)"

Line 347 (new 360): Space was deleted.

Line 357 (new 369): Dot was deleted.

Line 375 (new 388): "Some enzymes and vitamins in maca root can create an appetizing effect.." sentence was removed from the conclusion section.

Line 379 (new 392): "Thanks to the phenolic compounds in maca content, it is a plant with high antioxidant capacity" was omitted from the conclusion section.

Line 375-386 (new 388-401): The conclusion section has been modified and reshaped according to the reviewer's eligible opinion.

Reviewer 2 Report

The article intitled: “Influences of dietary supplementation with maca (Lepidium meyenii) on performance, parameters of growth curve, and carcass characteristics in Japanese quail by Firdevs Korkmaz Turgud and DoÄŸan Narinç” is very interesting and adds data on the use of Maca (Lepidium meyenii) also in this particular avian species, often used as an experimental model.

Below are my suggestions to the authors.

Introduction:

-On page 2 and line 53 add Tafuri et al., review 2021: Lepidium meyenii (Maca) in male reproduction. Tafuri S, Cocchia N, Vassetti A, Carotenuto D, Esposito L, Maruccio L, Avallone L, Ciani F. Nat Prod Res. 2021 Nov;35(22):4550-4559. doi: 10.1080/14786419.2019.1698572. Epub 2019 Dec 5.PMID: 31805775 Review.

-On page 2 and line 76: to explain the choice of the Japanese quail, the authors could add the following sentence “Among avian species, Japanese quail has been used in many sudies related to early sexual development and maturation (Maruccio et al., 2016, 2018), but also for the high production of the eggs and their regular deposition (Rodler and Sinovwatz, 2011).

Maruccio L, Lucini C, de Girolamo P, Avallone L, Solcan C, Nechita LE, Castaldo L. 2018. Neurotrophins and Trk receptors in the developing and adult ovary of Coturnix coturnix japonica. Ann Anat. 2018 Sep;219:35-43. doi: 10.1016/j.aanat.2018.04.008. Epub 2018 May 26.

Maruccio L, Castaldo L, D'Angelo L, Gatta C, Lucini C, Cotea C, Solcan C, Nechita EL. 2016. Neurotrophins and specific receptors in the oviduct tracts of Japanese quail (Coturnix coturnix japonica). Ann Anat. 2016 Sep;207:38-46. doi: 10.1016/j.aanat.2016.04.033. Epub 2016 May 7. PMID: 27167968

Rodler, D., Sinowatz. F. 2011. Immunohistochemical and ultrastructural characterization of the ovarian surface epithelium of Japanese quail (Coturnix japonica). Anim. Sci. J.  2(2):307-13. 

On page 2 and line 97: Unclear sentence. Can the authors explain this last sentence better?

On page 2 in Table 1 is reported: “Hesapta bulunan deÄŸerler” translete in English please

Materials and Methods:

-the dose of 0.05% and 0.1% is reported. Can the authors explain this choice?

- how many males and how many females were employed?

The Results have been clearly described also helped by numerous graphs.

Discussion

On page 9 and line 260: the authors wrote thatResearchers  reported that the addition of maca powder also did not affect feed consumption” …Can the authors add the reference and in which species does the report occur?

Conclusions

On page 11 and line 385: The authors are encouraged to pursue this research further

Author Response

Dear reviewer, thank you for your contribution, suggestions, and criticisms to the article. All corrections have been made and are shown below.

Line 53 (new 56): Tafuri et al (2021) is included both in the text and in the reference list.

Line 76 (new 79-82): The explanations were added.

Line 97 (new 107-109): In the study, feed consumption was determined as a group in brooder cages (from hatch-ing to 3 weeks of age). Then, it was determined individually in the rearing period (between 3-6 weeks of age).

Line 102 (new: 113): The expression in Table 1 has been translated into English.

Line 95: The doses of 0.05% and 0.1% were chosen according to the rates used in similar studies.

Line 128 (new 142): Added an expression indicating the number of males and females. "Fifty randomly selected quails from each experimental group and a total of 150 quails (71 females and 79 males) were sent to slaughter at the age of 42 days."

Line (new: 274-277): Here, the same work has been written twice by mistake, and the necessary correction has been made.

Reviewer 3 Report

Manuscript revision ID: animals-1499563 titled:

Influences of dietary supplementation with maca (Lepidium meyenii) on performance, parameters of growth curve, and carcass characteristics in Japanese quail

Abstract

The application form is missing. Please specify what is recommended for poultry production.

Aim of study

Please provide details of the parameters that will be tested.

M&M

Please provide the size of the experimental groups and the number of repetitions within each of them.

Is maca registered as a feed additive? If so, please provide the place of indexing / registration.

Table 1. text in Turkish - please correct

DCP - expand abbreviation,

Table 2 – expand abbreviations

L143 – please provide details of ” some reproductive characteristics”

L276-279 – This is a statistical analysis of the data, not the actual results obtained from a scientific experiment. Please review the literature for the analysis of production increments and effects, not for the analysis of statistical parameters..

L326-345 – Please explain the mechanism of action of maca and its active substances in the body of birds, and not only report the results of research by other scientists.

Conclusion

Too extensive. Please shorten it significantly, provide a synthetic overview of the current state of knowledge and what are the trends for the future. Provides general information. Please limit yourself to the results of your own research and formulate practical conclusions. What levels of maca are recommended for poultry production? Or…. is it not recommended for production at all?? Please specify your recommendations.

Author Response

Dear reviewer, thank you for your contribution, suggestions, and criticisms to the article. All corrections have been made and are shown below.

Line 27 (new 27): A recommendation statement about the results of the study has been added to the abstract section. (It may be advisable to add 0.1% maca powder to the diets of breeders.)

Line 23-30: In the abstract section, the details of the features that were tested for hypothesis were added. At the same time, the P value for H1 acceptance states is also added.

Line 96: The total number of birds used in the study is 300 (indicated on Line 92). In the rearing period, 100 quails were used in each experimental group, this information is added to line 96. At the end of the fattening period, 50 quails from each group were slaughtered and the numbers of females and males were added to line 145. In the reproduction period, 27 female and 9 male quails from each experimental group were used (In line 159). Measurements performed throughout the entire experiment were made individually using wing numbers. For this reason, each individual is considered as a replicate in such trials.

Answer: Is maca registered as a feed additive? If so, please provide the place of indexing / registration.

Question: Maca does not have a specific registration as a feed additive.

Line 116: The non-English text in Table 1 has been corrected.

Line 116: DCP - abbreviation was expanded

Line 137: Removed abbreviations for inflection point coordinates from Table 2.

Line 143 (new 161): Explanations were added for reproductive traits; "... determine egg production and some reproductive characteristics (fertility, embryonic mortalities, chick quality)."

Line 276-279 (new 296-298) The following statement has been removed from the aforementioned section; "In the study, it was determined that the addition of maca powder to the rations of Japanese quail did not have a statistical effect on the β0 and β1 parameters of the Gompertz model." In other parts of paper, some of the conclusions of statistical results were removed at the request of the reviewer.

Line 326: An analysis of the active components of the maca plant was not performed in this study. Instead, the effects of maca supplementation in quail diets on some yield-related phenotypic traits were studied. For this reason, discussions were made with the results of researchers who conducted similar studies. With the recommendation of another reviewer, the information on the mechanism of action related to maca in poultry was also removed from the paper.

Line 397: Considering the valuable suggestion of the reviewer, from the conclusion section "Some enzymes and vitamins in maca root can create an appetizing effect, which may explain the increase in feed consumption by adding maca to the ration." and "Thanks to the phenolic compounds in maca content, it is a plant with high antioxidant capacity." sentences have been deleted. In addition, a recommendation sentence has been added (It may be advisable to add 0.1% maca powder to the diets of breeder flocks).

In addition, a control of the article in terms of English language was made by a native speaker.